# `simple_rl`: Reproducible Reinforcement Learning in Python

**David Abel**
Brown University
Providence, RI 02903
david_abel@brown.edu

## Abstract

Conducting reinforcement-learning experiments can be a complex and timely process. A full experimental pipeline will typically consist of a simulation of an environment, an implementation of one or many learning algorithms, a variety of additional components designed to facilitate the agent-environment interplay, and any requisite analysis, plotting, and logging thereof. In light of this complexity, this paper introduces `simple_rl`[1], a new open source library for carrying out reinforcement learning experiments in Python 2 and 3 with a focus on simplicity. The goal of `simple_rl` is to support seamless, reproducible methods for running reinforcement learning experiments. This paper gives an overview of the core design philosophy of the package, how it differs from existing libraries, and showcases its central features.

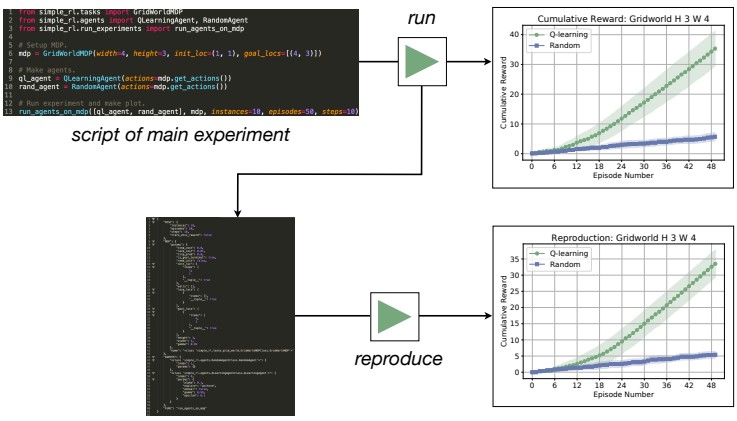

Figure 1: The core functionality of `simple_rl`: Create agents and an MDP, then run and plot their resulting interactions. Running an experiment also creates an experiment log (stored as a JSON file), which can be used to rerun the exact same experiment, thereby facilitating simple reproduction of results. All practitioners need to do, in theory, is share a copy of the experiment file to someone with the library to ensure result reproduction.

---

[1]https://github.com/david-abel/simple_rl

```
1  from simple_rl.agents import QLearningAgent, RandomAgent
2  from simple_rl.tasks import GridWorldMDP
3  from simple_rl.run_experiments import run_agents_on_mdp
4
5  # Setup MDP.
6  mdp = GridWorldMDP(width=4, height=3, init_loc=(1, 1), goal_locs=[(4, 3)])
7
8  # Make agents.
9  ql_agent = QLearningAgent(actions=mdp.get_actions())
10 rand_agent = RandomAgent(actions=mdp.get_actions())
11
12 # Run experiment and make plot.
13 run_agents_on_mdp([ql_agent, rand_agent], mdp, instances=5, episodes=50,
       steps=10)
```

Figure 2: Example code for running a basic experiment. First, define a grid-world MDP (line 6), then make our agents (line 9-10), and then run the experiment (line 13). Running the above will generate the plot shown in Figure 4.

# 1   Introduction

Reinforcement learning (RL) has recently soared in popularity due in large part to recent success in challenging domains, including learning to play Atari games from image input [27], beating the world champion in Go [32], and robotic control from high dimensional sensors [21]. In concert with the field's growth, experiments have become more complex, leading to new challenges for empirical evaluation of RL methods. Recent work by Henderson et al. [16] highlighted many of the issues involved with handling this new complexity, raising concerns about emerging RL experimental practices. Additionally, Python has become a prominent programming language used by machine-learning researchers due to the availability of powerful deep learning libraries like PyTorch [29] and tensorflow [1], along with scipy [19] and numpy [28].

To accommodate this growth, there is a need for a simple, lightweight library that supports quick execution and analysis of RL experiments in Python. Certainly, many libraries already fulfill this need for many uses cases—-as will be discussed in Section 2, many effective RL libraries for Python already exist. However, the design philosophy and ultimate end user of these packages is distinct from those targeted by simple_rl: those users who seek to quickly run simple experiments, look at a plot that summarizes results, and allow for the quick sharing and reproduction of these findings.

The core design principle of simple_rl is that of *simplicity*, per its name. The library is stripped down to the bare necessities required to run basic RL experiments. The focus of the library is on traditional, tabular domains, though it does have the capacity to cooperate with high-dimensional environments like those offered by the OpenAI Gym [6]. The assumed objective of a practitioner using the library is to define (1) an RL agent (or collection of agents), (2) an environment (an MDP, POMDP, or similar Markov model), (3) let the agent(s) interact with the environment, and (4) view and analyze the results of this interaction. This basic pipeline serves as the "end-game" of simple_rl, and dictates much of the design and its core features. A block diagram of this process is presented in Figure 1: run an experiment, see the results, and reproduce these results according to an auto-generated JSON file logging the experimental details. The actual code of the experiment run is shown in Figure 2: in around five lines, we define a $Q$-Learning instance, a random actor, and a simple grid-world domain, and let these agents interact with the environment for a set number of instances. As mentioned, running this code produces both a JSON file tracking the experiment that can be used (or shared) to run the same experiment again, and regenerate the plot seen in Figure 4a.

# 2   Relation To Other Libraries

Many excellent libraries already exist in Python for carrying out RL experiments. What separates simple_rl? As the name suggests, its distinguishing feature is its emphasis on simplicity, which also brings a shortage of certain features. We here describe the objectives of other RL libraries in

Python, and briefly cover what some have implemented in case those are a better fit for the needs of different programmers.

## 2.1 RLPy

RLPy offers a well documented, expansive library for RL and planning experiments in Python 2 [15]. The library includes a similar overall structure to that of `simple_rl`: the core entities are agents, environments, experiments, policies, and representations. The main focus of RLPy is on value-function approximation, but the library also offers several MDP solvers in the form of the usual dynamic programming algorithms like value iteration [4] and policy iteration [18]. Notably, the library also includes a large number of canonical RL tasks, including Mountain Car, Acrobot, Puddle World, Swimmer, and Cart Pole.

Get it here: https://github.com/rlpy/rlpy

## 2.2 mushroom

Mushroom is a new library aimed at simplifying RL experimentation with OpenAI gym and tensorflow, but also offers support for traditional tabular experiments [13]. Mushroom offers implementations of many recent Deep RL algorithms, including DQN [27], Stochastic Actor-Critic [12], and a template for Policy Gradient algorithms. All of its neural network code is based on tensorflow. Additionally, Mushroom comes with noteworthy RL tasks like Mountain Car, Inverted Pendulum, and a classic Linear-Quadratic Regulator control task.

Get it here: https://github.com/AIRLab-POLIMI/mushroom

## 2.3 PyBrain

PyBrain is an established, expansive, general purpose library for machine learning in Python [30], but also offers infrastructure for conducting RL experiments with a similar focus to RLPy. The library includes a number of the standard environments and agents, with a large number of model-free algorithms.

Get it here: http://www.pybrain.org/

## 2.4 keras-rl

`keras-rl` provides integration between Keras [9] and many popular Deep RL algorithms. `keras-rl` offers an expansive list of implemented Deep RL algorithms in one place, including: DQN, Double DQN [37], Deep Deterministic Policy Gradient [23], and Dueling DQN [38]. For those that use Keras for deep learning and mostly want to focus on deep RL, `keras-rl` library is a great choice.

Get it here: https://github.com/keras-rl/keras-rl

## 2.5 python-rl

`python-rl` [11] provides integration with the classic language-agnostic framework RL-Glue [36]. The main goal of this library is to bring RL-Glue up to date with a few somewhat more recent features, agents, and environments in common RL experiments.

Get it here: https://github.com/amarack/python-rl

## 2.6 reinforcement-learning

`reinforcement-learning` offers an excellent resource for RL education—it is designed to be paired with David Silver's online RL course[2] [5]. The library contains many central algorithms, including value iteration, policy iteration, Q-Learning [39], SARSA [33], and Pol-

---

[2] https://www.youtube.com/watch?v=2pWv7GOvuf0

icy Gradient [40, 35]. Programmers planning to go through David Silver's course may find the `reinforcement-learning` library the most suitable package.

Get it here: https://github.com/dennybritz/reinforcement-learning

## 2.7 `dopamine`

`dopamine` is a recently released library [3] offering many of the most recent deep RL algorithms including Rainbow [17], Prioritized Experience Replay [31], and Distributional RL [2], with an eye for reproducibility in the ALE based on the suggestions given by [25]. `dopamine` offers a lot for people whose main agenda is to run experiments in the ALE or perform new research in deep RL.

Get it here: https://github.com/google/dopamine

.........................

To summarize: Many great packages are already out there. The main differentiating features of `simple_rl` are (1) quick generation of plots, (2) focus on reproducibility, and (3) emphasis on simplicity, both in terms of algorithmic development and its attachment to classical RL problems (like grid worlds).

# 3 Overview of Features

We begin by unpacking the example in Figure 2 to showcase the main design philosophy of `simple_rl`.

## 3.1 The Core: Agents and MDPs

The library primarily consists of *agents* and *environments* (called "tasks" in the library).

Agents, by default, are all subclasses of the abstract class, `Agent`, which is only responsible for a method `act(self, state, reward)` $\rightarrow$ `action`. A list of agents, planning algorithms, and tasks currently implemented is presented in Table 1.

Tasks, for the most part, all inherit from the abstract MDP class, `MDP`. The core of an MDP is its transition function and reward function, captured in the abstract class by class-wide variables, `transition_func` and `reward_func`:

$$\text{transition\_func(state, action)} \rightarrow \text{state,} \tag{1}$$

$$\text{reward\_func(state, action)} \rightarrow \text{reward.} \tag{2}$$

When defining an MDP instance, you must pass in functions of $T$ and $R$ that *output* a state and reward, respectively. In this way, no MDP is ever responsible for enumerating either $\mathcal{S}$ or $\mathcal{A}$ explicitly, thereby allowing for (1) simple specification of these two functions, and (2) efficient implementation of high-dimensional domains—we need only represent and store the states that are visited during experimentation.

Naturally, MDP subclasses have a variety of arguments—in the earlier grid-world example, we saw the `GridWorldMDP` class take as input the dimensions of the grid, a starting location, and a list of goal locations. Such inputs are typical to MDP classes in `simple_rl`.

### 3.1.1 Running Simple Experiments

Defining an agent and an MDP is almost all that is needed to run an experiment. The final component required is an experiment function from the `run_experiments.py` file. This file contains a number of different experiment types that are catered to the different environment types (POMDPs, Markov Games, and so on). For now, let us focus on `run_agents_on_mdp` function, which is the most canonical. As per the example in Figure 2, this function takes as input at minimum a list of agents and an MDP instance. A user can also specify experimental parameters like `instances`, `episodes`, and `steps`, which indicate the following:

- `instances`: The number of times to repeat the entire experiment (will be used to form 95% confidence intervals for all experiments conducted).

| | |
|---|---|
| *RL Agents* | Q-Learning, RMax, DelayedQ, DoubleQ, Random, Fixed |
| | Linear Q-Learning, DQN, LinUCB. |
| *Planning Algorithms* | Value Iteration, Bounded RTDP, MCTS |
| | |
| *MDPs* | Chain, Grid World, Randomized Graph, Open AI Gym |
| | Combo Lock, Puddle, Hanoi, Bandit |
| *OOMDPs* | Taxi, Trench, Cleanup |
| *POMDPs* | Maze |
| *Markov Games* | Grid Games, Rock Paper Scissors, Prisoner's Dilemma, Gather |

Table 1: An overview of Agents and MDPs in `simple_rl`.

```python
1   from simple_rl.tasks import GymMDP
2   from simple_rl.agents import RandomAgent, LinearQAgent
3   from simple_rl.run_experiments import run_agents_on_mdp
4
5   # Gym MDP
6   gym_mdp = GymMDP(env_name='CartPole-v1', render=True)
7   num_feats = gym_mdp.get_num_state_feats()
8
9   # Setup agents and run.
10  rand_agent = RandomAgent(gym_mdp.get_actions())
11  lin_q_agent = LinearQAgent(gym_mdp.get_actions(), num_feats, rbf=True)
12  agents = [lin_q_agent, rand_agent]
13
14  # Run.
15  run_agents_on_mdp(agents, gym_mdp, instances=5, episodes=5000, steps=200)
```

Figure 3: Running experiments in the OpenAI Gym.

- episodes: The number of *episodes* per instance. An episode will consist of steps number of steps, after which the agent is reset to the start state (but gets to remember what it has learned so far).

- steps: The number of steps per episode.

The plotting is set up to plot all of the above appropriately. For instance, if a user sets episodes=1 but steps=50, then the library produces a step-wise plot (that is, the x-axis is steps, not episodes).

Running the function run_agents_on_mdp will create a JSON file detailing all of the components of the experiment needed to rerun it. Then, it will create a folder locally, "results", store each agent's stream of received rewards, and print out the status of the experiment to console. When the experiment concludes, a learning curve with 95% confidence intervals will be generated (via simple_rl/utils/chart_utils.py and opened. The JSON file lets users of the library reconstruct and rerun the original experiment using another function from the run_experiments.py script. In this way, the JSON file is effectively a certificate that this plot can be reproduced if the same experiment were run again. We provide more detail on this feature in Section 3.2.

We can also run a similar experiment in the OpenAI Gym (Figure 3).

As can be seen in Figure 3, the structure of the experiment is identical. Since we define a GymMDP, we pass as input the name of the environment we'd like to produce: In this case, we're running experiments in CartPole-v1, but any of the usual Gym environment names will work. We can also pass in the render boolean flag, indicating whether or not we'd like to visualize the learning process. Alternatively, we can pass in the render_every_n_episodes flag (along with render=True), which will only render the agent's learning process every $N$ episodes.

On longer experiments, we may want additional feedback about the learning process. For this purpose, the `run_agents_on_mdp` function also takes as input a Boolean flag `verbose`, which, if true, will provide detailed episode-by-episode tracking of the progress of the experiment to the console. There are a number of other ways to run experiments, but these examples capture the core experimental cycle.

**Other Environment Types**    The library offers support for other types of environments beyond typical MDPs, including classes for Object-Oriented MDPs or OOMDPs [14], $k$-Armed Bandits [8], Partially Observable MDPs or POMDPs [20], a probability distribution over MDPs for lifelong learning [7], and Markov Games [24]. Aspects of these classes are handled slightly differently to accommodate the different kinds of decision-making problems they capture, but the interface to run experiments with each type is nearly identical. Examples for how to run experiments with each type of environment are included in the examples directory in the repository along with a test script that ensures each example can run on a given machine. Running experiments with these other environment types is the same as the pipeline so far described: a function in the `run_experiments.py` script will handle all of the interactions between agent(s) and environment and produce a plot when the experiment finishes. Notably, the reproducibility feature is not yet fully developed for all environment types. This is a major direction for future development of the library.

## 3.2   Reproducibility

Due to its simplicity, the library is naturally suited for reproducing results from previously run experiments. As mentioned, *every* experiment that is conducted using the library will create a directory with the experiment name containing a JSON file "`full_experiment_data.json`" that enumerates every parameter, agent, MDP, and type needed to launch the exact same experiment another time. The idea is that these files can be shared across users of the library—if a user gives someone else this file (and the necessary agents and environments), it is a contract that they can rerun *exactly* the same experiment just run using `simple_rl`.

Using one of these experiment files, the function `reproduce_from_exp_file(exp_name)`, will read the experiment file, reconstruct all the necessary components, rerun the entire experiment, and remake the plot. Thus, providing one of these JSON files is to be interpreted as a certificate that this experiment is guaranteed to produce similar results.

As an example, consider again the code from Figure 2. Running this code will create: (1) the "`results`" directory, (2) the "`gridworld_h-3_w-4`" directory within `results`, and (3) the "`full_experiment_data.json` file, which contains *all* necessary parameters to rerun the experiment.

Suppose someone provided the directory `gridworld_h-3_w-4` containing the experiment file for the above grid-world experiment. Then, we could run the following code:

```
1    from simple_rl.run_experiments import reproduce_from_exp_file
2
3    reproduce_from_exp_file("gridworld_h-3_w-4")
```

Which will automatically generate the plot in Figure 4b.

To ensure reproducibility of new subclasses or other bells and whistles attached to the library, any agent or MDP must implement the "`get_parameters(self)`" method that returns a dictionary containing all relevant parameters for the instance to be reconstructed. For example, consider the `QLearningAgent` class in Figure 5.

Any introduced subclass that wants to play along well with the reproduction infrastructure in `simple_rl` must have such a method.

We stipulate that this is a lightweight means of ensuring reproduction for three reasons: 1) it is entirely obfuscated from the programmer, as all tracking of experimental parameters is done automatically, 2) a single, universally formatted document (JSON) contains all the information needed to guarantee reproduction of results (along with a copy of the library itself, and any new agents/MDPs), and 3) the library is simple enough that most experiments consist of only a small number of moving

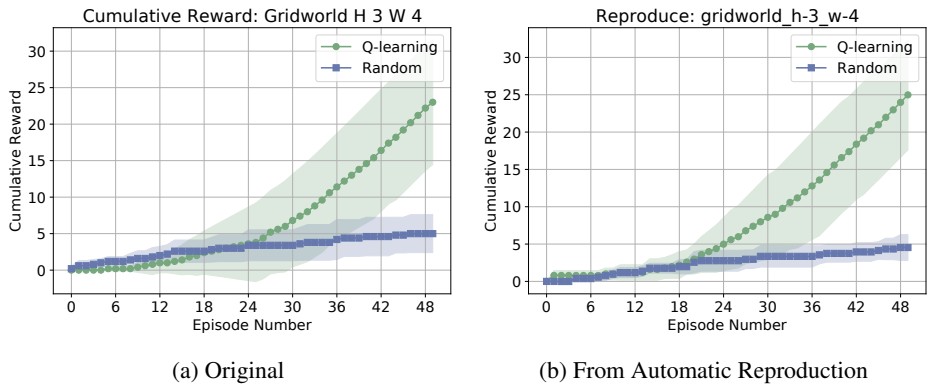

(a) Original        (b) From Automatic Reproduction

Figure 4: Original results (left) and results generated by reproducing the experiment (right).

```python
1   def get_parameters(self):
2       '''
3       Returns:
4           (dict) key=param_name (str) --> val=param_val (object).
5       '''
6       param_dict = defaultdict(int)
7
8       param_dict["alpha"] = self.alpha
9       param_dict["gamma"] = self.gamma
10      param_dict["epsilon"] = self.epsilon_init
11      param_dict["anneal"] = self.anneal
12      param_dict["explore"] = self.explore
13
14      return param_dict
```

Figure 5: The get_parameters method of QLearningAgent.

parts. The feature to reproduce from a JSON does not yet fully support all environment types, but it is an active area of development for the library.

To recap, the introduced components define the essence of the library:

- Center everything around *agents*, *MDPs*, and interactions thereof.
- Completely obscure the complexity of plotting and experiment tracking from the programmer, while making it simple to plot and reproduce results if needed.
- Simplicity above all else.
- Treat things *generatively*—namely, MDPs transition models and reward functions are best implemented as functions that return a state or reward, rather than enumerate all state–actions pairs.

## 3.3 Utilities

In addition to the core experimental pipeline described above, the library is well stocked with other utilities useful for RL and planning.

**Plotting** As is shown by Figure 1, plotting is tightly coupled with running experiments. Each experiment type is connected with the same plotting script, stored in the library in utils/chart_utils.py. The basic plot shows some measure of time along the x-axis (either in episodes run or steps taken), with cumulative reward shown in the y-axis for each given algorithm. While this plot is the default learning curve generated, the experimental pipeline gives the end programmer control over the type of plot generated. First, the flag cumulative_plot for all of the core

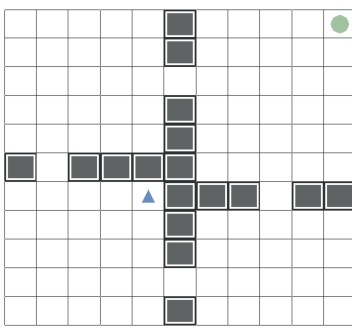

Figure 6: Example visual generated by the library

experiment functions is set to `True` by default (as in `run_agents_on_mdp`, `run_agents_lifelong`). Thus, if we simply run the experiment with this flag set to `False`, we'll produce an average reward plot instead. Second, the default y-axis is cumulative *reward*—sometimes, though, we'd like to measure the *discounted* reward acquired by the agent. To do so, we set the `track_disc_reward` flag of any of the core experimental functions to `True`. There are also mechanisms for plotting the wall-clock time taken by each agent, and plotting the percentage of successful runs of each agent, where success is defined according to a user defined function on the reward stream received by the agent. For more details on plotting, see the `chart_utils.py` script.

**Visuals**   The library offers bare bones visuals for the grid world domains using `pygame`[3]. An example is presented in Figure 6; in this case, the learning process is visualized while the experiment runs. The library also supports visualizing policies and value functions, so long as an MDP comes along with a `draw_state` method, and an interactive mode where the user can control the agent via keyboard input. However, visuals are very much an underdeveloped aspect of the library. A major point of future development is to equip `simple_rl` with a comprehensive suite of visualization and analysis tools.

**Abstraction**   A core approach to RL involves forming *abstractions*, either of state [22] or action [34]. `simple_rl` contains support for planning or learning with either state aggregation functions, which compress a given MDP's state space into a smaller one, and options, which encode long horizon sequences of actions, useful for targeted exploration and efficient planning.

**Planning**   The library includes several default planning algorithms such as Value Iteration, Monte Carlo Tree Search [10], and Bounded Real Time Dynamic Programming [26]. Planners can be used to compute the value function, the optimal (or near-optimal) policy, or enumerate a state-action space (see `planning_example.py` in the repository).

## 4   Conclusion

`simple_rl` offers a lightweight suite of tools for conducting RL experiments in Python 2 and 3. Its design philosophy focuses on obfuscating complexity from the end user, including the tracking of experimental details, generation of plots, and construction of agents and MDPs. This leads to a package that is relatively light in features but comes with an ease of use that lets only a few lines of code generate learning curves that are guaranteed to be reproducible. The library is available on the Python package index, and thus can be installed with the usual `pip install simple_rl`. In progress documentation is available as well.[4] Many features are currently under development: the most important near term goal is to expand the suite of reproducibility tools to account for more variety across different operating systems and other variables that might impact experiments. Additionally, the library lacks a suite of basic deep RL algorithms for use in experimentation, a general interface for visualizing MDPs (and other environments), and a more expansive collection of tasks, RL algorithms, and planning algorithms.

---

[3] https://pygame.org
[4] https://david-abel.github.io/simple_rl/docs/index.html

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
