# OpenReview forum: "simple_rl: Reproducible Reinforcement Learning in Python"
_ICLR.cc/2019/Workshop/RML — RML 2019_

### Official Review · AnonReviewer1 · 2019-04-01
**Interesting Framework**

**Rating:** 3
**Confidence:** 3

**Review:**

Summary: This paper discusses some features of simple_rl, a framework for RL in Python that emphasizes simplicity and tools for reproducibility.  This is a nice workshop paper but would benefit from a clearer discussion of the related work.

Notes:
  -SimpleRL is an algorithm for RL experiments in Python.
  -After creating agents and MDPs, an experiment log as a json is produced.
  -Practitioners can share a copy of the experiment file to ensure reproducibility.  However with docker or containers this should always be achievable, unless there’s some guarantee that all of the seeds are in the experiment file?
  -Consists of MDP objects and agent objects.
  -Main design goal is simplicity.
  -MDP has “transition function” and “reward function” objects.  I wonder how well the structure generalizes to model-based RL?
  -Some utilities for reproducing results from the json files.
  -Plotting utilities are included.

Comments:
  -Section 2 could do a better job of making it clearer how the simplicity of simple_rl isn’t achieved by the other libraries.  Nonetheless, it’s still a nice overview.

---

### Decision · Program_Chairs · 2019-04-05
**Acceptance Decision**

Accept